Genomic characterization and phylogenetic analysis of Salmonella enterica serovar Javiana

Hudson Lauren K. 1
Constantine-Renna Lisha 2
Thomas Linda 3
Moore Christina 3
Qian Xiaorong 3
Garman Katie 2
Dunn John R. 2
Denes Thomas G. tdenes@utk.edu 1
1 Department of Food Science, University of Tennessee , Knoxville , TN , United States of America
2 Tennessee Department of Health , Nashville , TN , United States of America
3 Division of Laboratory Services, Tennessee Department of Health , Nashville , TN , United States of America
Mossong Joël
Electronic publication date: 2020 Nov 20
Publication date: 2020
Volume: 8
Electronic Location ID: e10256
Received 2020 Jun 9; Accepted 2020 Oct 6
Copyright: ©2020 Hudson et al.
Copyright year: 2020
Copyright holder: Hudson et al.
License: This is an open access article distributed under the terms of the Creative Commons Attribution License, which permits unrestricted use, distribution, reproduction and adaptation in any medium and for any purpose provided that it is properly attributed. For attribution, the original author(s), title, publication source (PeerJ) and either DOI or URL of the article must be cited.
License URL: https://creativecommons.org/licenses/by/4.0/

Keywords: Salmonella Javiana, Pan-genome-wide association study

Funding: Epidemiology and Laboratory Capacity for Infectious Diseases (ELC) NU50CK000528-01 This work was supported by Epidemiology and Laboratory Capacity for Infectious Diseases (ELC) grant 6 NU50CK000528-01 funded by the Centers for Disease Control and Prevention (CDC) and multistate project S1077 “Enhancing Microbial Food Safety by Risk Analysis.” The funders had no role in study design, data collection and analysis, decision to publish, or preparation of the manuscript.

==============================
Salmonella enterica serovar Javiana is the fourth most reported serovar of laboratory-confirmed human Salmonella infections in the U.S. and in Tennessee (TN). Although Salmonella ser. Javiana is a common cause of human infection, the majority of cases are sporadic in nature rather than outbreak-associated. To better understand Salmonella ser. Javiana microbial population structure in TN, we completed a phylogenetic analysis of 111 Salmonella ser. Javiana clinical isolates from TN collected from Jan. 2017 to Oct. 2018. We identified mobile genetic elements and genes known to confer antibiotic resistance present in the isolates, and performed a pan-genome-wide association study (pan-GWAS) to compare gene content between clades identified in this study. The population structure of TN Salmonella ser. Javiana clinical isolates consisted of three genetic clades: TN clade I (n = 54), TN clade II (n = 4), and TN clade III (n = 48). Using a 5, 10, and 25 hqSNP distance threshold for cluster identification, nine, 12, and 10 potential epidemiologically-relevant clusters were identified, respectively. The majority of genes that were found to be over-represented in specific clades were located in mobile genetic element (MGE) regions, including genes encoding integrases and phage structures (91.5%). Additionally, a large portion of the over-represented genes from TN clade II (44.9%) were located on an 87.5 kb plasmid containing genes encoding a toxin/antitoxin system (ccdAB). Additionally, we completed phylogenetic analyses of global Salmonella ser. Javiana datasets to gain a broader insight into the population structure of this serovar. We found that the global phylogeny consisted of three major clades (one of which all of the TN isolates belonged to) and two cgMLST eBurstGroups (ceBGs) and that the branch length between the two Salmonella ser. Javiana ceBGs (1,423 allelic differences) was comparable to those from other serovars that have been reported as polyphyletic (929–2,850 allelic differences). This study demonstrates the population structure of TN and global Salmonella ser. Javiana isolates, a clinically important Salmonella serovar and can provide guidance for phylogenetic cluster analyses for public health surveillance and response.

Introduction

Salmonella enterica subspecies enterica serovar Javiana (Salmonella ser. Javiana) was ranked the fourth most reported serovar (behind Enteritidis, Typhimurium, and Newport) in the United States in 2015, accounting for 7.4% (n = 575) of laboratory-confirmed human Salmonella infections (Centers for Disease Control and Prevention, 2017a). The incidence rate (IR) was 1.17 per 100,000 persons (Centers for Disease Control and Prevention, 2017a). In 2016, 2,719 culture-confirmed human Salmonella ser. Javiana infections were reported to the Laboratory-based Enteric Disease Surveillance (LEDS) system (9.8% of Salmonella infections; IR of 1.43 per 100,000 persons) (Centers for Disease Control and Prevention, 2018). The number of actual illnesses is likely higher according to CDC estimates of 29.3 actual cases per each laboratory-reported case (Scallan et al., 2011). Nationally, Salmonella ser. Javiana IR increased 136% from 2001 to 2016 and 325% since 1970 (Centers for Disease Control and Prevention, 2018). From a total of 2,390 Salmonella ser. Javiana cases in FoodNet states that occurred from 1996 to 2006, 20.6% resulted in hospitalization, 2.8% in invasive disease, and 0.4% in death (Jones et al., 2008). Infections in infants and young children occur at higher rates than other Salmonella serovars (Jones et al., 2008; Shaw et al., 2016; Srikantiah et al., 2004). Javiana is also the fourth most common clinically isolated Salmonella serovar in Tennessee (TN), accounting for 7.8% of culture-confirmed Salmonella infections in 2016 (Centers for Disease Control and Prevention, 2018). The IR for TN reported by LEDS in 2016 was 1.17 (per 100,000 population), with a total of 74 culture-confirmed human infections reported (Centers for Disease Control and Prevention, 2018).

Salmonella ser. Javiana is commonly found in the South Eastern US (Centers for Disease Control and Prevention, 2013; Reddy et al., 2016), particularly South Georgia (GA), Central Arkansas, and coastal areas of South and North Carolina (Srikantiah et al., 2004). States with the highest IR (per 100,000 population) of culture-confirmed human Salmonella ser. Javiana reported to LEDS in 2016 were Mississippi (6.65), South Dakota (6.29), South Carolina (5.17), North Carolina (3.50), and Georgia (3.50) (Centers for Disease Control and Prevention, 2018). Geographical restriction or distribution, as seen for Salmonella ser. Javiana, may indicate association with local food products, persistence in or adaptation to specific regional environments, and/or an environmental or animal reservoir with a specific geographic territory (Clarkson et al., 2010; Reddy et al., 2016; Strawn et al., 2014). Salmonella ser. Javiana cases are typically highest in summer (Clarkson et al., 2010; Reddy et al., 2016; Srikantiah et al., 2004), suggesting higher populations of an animal reservoir, higher exposure to sources or reservoirs, and/or increased contamination levels (Boore et al., 2015).

Due to the geographical distribution of this serovar, some researchers have evaluated correlations between incidence and environmental conditions or potential contamination sources. Huang et al. (2017) described the association between wetlands and incidence of Salmonella ser. Javiana. They found that freshwater forested/scrub-shrub wetland and freshwater emergent wetland were both significantly associated with increased Salmonella ser. Javiana IR in GA, Maryland (MD), and TN. Additionally, freshwater pond wetlands were both significantly associated with increased IR in GA and TN. Shaw et al. (2016) found a statistically significant association between increased rates of Salmonella ser. Javiana infection and percentage of housing in rural areas in Georgia. Furthermore, they found a statistically significant association between increased Salmonella ser. Javiana infection rates and the presence of broiler feeding operations in MD (Shaw et al., 2016). Rural areas in GA and MD have a high density of broiler chicken operations and rural areas in TN have a high density of both broiler chicken and cattle operations (Shaw et al., 2016). The higher presence of these operations in rural areas could facilitate transmission of Salmonella ser. Javiana and other serovars in a variety of ways. As these operations are found in high density in these rural areas, they may employ a large number of residents in the area. These employees may be directly exposed to Salmonella occupationally and indirectly expose others in those communities via items like clothes and shoes (Shaw et al., 2016). The high density of these operations could also lead to environmental transmission via contamination of groundwater and surface water with untreated animal waste (Shaw et al., 2016). Shaw et al. (2016) did not find any statistically significant correlations between rurality or presence of broiler, cattle, dairy, or hog operations and IR ratios of Salmonella ser. Javiana in TN.

Salmonella ser. Javiana outbreaks have been linked to chicken (Jackson et al., 2013), pork (Jackson et al., 2013), cheese (Alley & Pijoan, 1942; Hedberg et al., 1992), shrimp (Venkat et al., 2018), produce (Bennett et al., 2015; Blostein, 1993; Centers for Disease Control and Prevention, 2005; Centers for Disease Control and Prevention, 2007; Hanning, Nutt & Ricke, 2009; Jackson et al., 2013; Sandt et al., 2006; Sivapalasingam et al., 2004; Srikantiah et al., 2005; Toth et al., 2002; Walsh et al., 2014), spices (Lehmacher, Bockemühl & Aleksic, 1995; Zweifel & Stephan, 2012), ill foodhandlers (Elward et al., 2006; Lee, Peppe & George, 1998), and contact with amphibians (Srikantiah et al., 2004). According to the National Outbreak Reporting System (NORS), there have been eight Salmonella ser. Javiana outbreaks involving TN, five multistate and three single-state and all were foodborne. Identified vehicles included tomatoes, cucumbers, tilapia, fajita (beef), and iceberg lettuce. All of the TN outbreaks were in restaurant settings.

Salmonella ser. Javiana has been isolated from a variety of foods, including seafood (Mezal, Stefanova & Khan, 2013), white pepper (Mezal, Stefanova & Khan, 2013), produce (Duffy et al., 2005; Elviss et al., 2009; Mezal, Stefanova & Khan, 2013; Reddy et al., 2016), and pecans (Brar, Strawn & Danyluk, 2016). Environmentally, Salmonella ser. Javiana has been isolated from surface water and sediment (Bell et al., 2015; Li et al., 2014; Micallef et al., 2012), poultry farms (Gama, Berchieri Jr & Fernandes, 2003; Rodriguez et al., 2006; Santos et al., 2007), dairy and livestock farms (Adesiyun et al., 1996; Oliveira et al., 2002; Rodriguez et al., 2006), irrigation water (Duffy et al., 2005), and packing shed equipment surfaces (Duffy et al., 2005). It has also been recovered from wildlife (Drake et al., 2013; Gruszynski et al., 2014; Lockhart et al., 2008; Miller et al., 2014), pets (Adesiyun, Campbell & Kaminjolo, 1997; Leahy et al., 2016; Seepersadsingh, Adesiyun & Seebaransingh, 2004; Woodward, Khakhria & Johnson, 1997), and zoo animals (Gopee, Adesiyun & Caesar, 2000). The diversity of animals found carrying Salmonella ser. Javiana includes amphibians (Drake et al., 2013), reptiles (Lockhart et al., 2008; Woodward, Khakhria & Johnson, 1997), birds (Gopee, Adesiyun & Caesar, 2000; Gruszynski et al., 2014), and mammals (Adesiyun, Campbell & Kaminjolo, 1997; Gopee, Adesiyun & Caesar, 2000; Gruszynski et al., 2014; Iveson & Bradshaw, 1973; Leahy et al., 2016; Miller et al., 2014; Seepersadsingh, Adesiyun & Seebaransingh, 2004). As Salmonella ser. Javiana has been isolated from and associated with contact with reptiles and amphibians, this may play a role in contamination of plant-based food commodities (Centers for Disease Control and Prevention, 2002; Clarkson et al., 2010; Jackson et al., 2013). A recently published systematic review identified the following risk factors associated with Salmonella ser. Javiana infection: consumption of fresh produce (tomatoes and watermelons), herbs (paprika-spice), dairy products (cheese), drinking contaminated well water, and animal contact (Mukherjee et al., 2019). Clarkson et al. (2010) found consumption of well water, reptile/amphibian contact, and exposure to recreational water associated with Salmonella ser. Javiana infection in GA and TN, but found consumption of tomatoes and poultry protective.

Though antibiotics are generally not used to treat uncomplicated Salmonella infections, when necessary, antibiotics most commonly used include ampicillin (penicillin), chloramphenicol (phenicol), ciprofloxacin (fluoroquinolone), ceftriaxone (cephalosporin), trimethoprim-sulfamethoxazole (folate pathway inhibitor, sulfonamide), amoxicillin (penicillin), and azithromycin (macrolide) (Cuypers et al., 2018; Eng et al., 2015; Gilbert et al., 2016; Jajere, 2019; Shane et al., 2017). The 2019 “Antibiotic Resistance Threats in the United States” report lists drug-resistant nontyphoidal Salmonella as a “serious threat” and details increasing numbers of isolates ciprofloxacin nonsusceptible, ceftriaxone resistant, or with decreased susceptibility to azithromycin (Centers for Disease Control and Prevention, 2019a). From the National Antimicrobial Resistance Monitoring System (NARMS) Now Salmonella ser. Javiana human isolate data from 1996-2019 (Centers for Disease Control and Prevention, 2019b), the highest prevalences of phenotypic antibiotic resistance were to streptomycin (2.27%; aminoglycoside), ampicillin (1.39%), and tetracycline (1.05%). Resistance to amoxicillin-clavulanic acid, cefoxitin (cephalosporin), ceftiofur (cephalosporin), ceftriaxone, cephalothin (cephalosporin), chloramphenicol, sulfamethoxazole/sulfisoxazole (sulfonamides), and trimethoprim-sulfamethoxazole were all less than 1% (Centers for Disease Control and Prevention, 2019b). Resistance to azithromycin or ciprofloxacin was not reported (Centers for Disease Control and Prevention, 2019b). Resistance among Salmonella ser. Javiana isolates may be lower due to the association with wild animal (e.g., reptile and amphibian) and other environmental reservoirs in contrast to food animal-associated serovars.

Though Salmonella ser. Javiana is a prevalent serovar in both the US and TN, little is known about the genomic population structure. The objectives of this study were to retrospectively study isolates of Salmonella ser. Javiana from patients in TN in 2017-2018 in order to identify epidemiologically-relevant trends, determine the genomic population structure, and describe the defining genomic features of TN clades. Additionally, we studied expanded datasets representing global diversity to determine the overall population structure of Salmonella ser. Javiana and to compare it to other Salmonella serovars.

Materials & Methods

Sequencing, preprocessing, and genome assembly of TN isolates

BioSample numbers and metadata for Salmonella ser. Javiana (n = 111) isolates from patients in TN from January 2017 through October 2018 were obtained from the Tennessee Department of Health (TDH) (Data S1). Tennessee population data (2018) used for calculating incidence rates (IR) was obtained from the U.S. Census Bureau (U.S. Census Bureau, 2020) and IR per county were mapped using Tableau Desktop Public Edition (v2019.2.1) (Tableau Software, 2019). PFGE and whole-genome sequencing were performed by the TDH Division of Laboratory Services according to PulseNet protocols (Centers for Disease Control and Prevention, 2016; Centers for Disease Control and Prevention, 2017b). For PFGE, XbaI was used as the primary restriction enzyme. Genomic DNA was extracted using Qiagen DNeasy Blood & Tissue kits, libraries were prepared using Nextera XT kits, and sequencing was performed on an Illumina MiSeq platform using Illumina MiSeq v2 chemistry (500 cycle) to produce 250bp paired-end reads. Raw reads were downloaded from the NCBI SRA database, trimmed using Trimmomatic v0.35 (Bolger, Lohse & Usadel, 2014) (with the following parameters: ILLUMINACLIP: NexteraPE-PE.fa:2:30:10 LEADING:3 TRAILING:3 SLIDINGWINDOW:4:15 MINLEN:36), and quality checked using FastQC v0.11.7 (Andrews, 2010) and MultiQC v1.5 (Ewels et al., 2016) to combine the results. The trimmed reads were assembled into contigs using SPAdes v3.12.0 (Bankevich et al., 2012) with the careful option. Assembly statistics were generated by BBMap v38.88 (Bushnell, 2018), SAMtools v0.1.8 (Li et al., 2009), and QUAST v4.6.3 (Gurevich et al., 2013). SeqSero (Zhang et al., 2015) was used to confirm serovar designations.

SNP detection and phylogenetic analyses

A reference-free SNP detection analysis was initially performed with the TN isolates to determine the overall population structure free of reference choice bias. The assemblies were analyzed using KSNP3.1 (Gardner, Slezak & Hall, 2015) and the resulting core SNP matrix fasta file was then used to construct a phylogenetic tree in Mega7 (Kumar, Stecher & Tamura, 2016) with 100 bootstrap replicates (Felsenstein, 1985). The evolutionary distances were computed using the number of differences method (Nei & Kumar, 2000) and the evolutionary history was inferred using the Neighbor-Joining method (Saitou & Nei, 1987). The final tree was visualized and annotated using iTOL (Letunic & Bork, 2016). Isolates that weren’t serovar Javiana (based on SeqSero results) and were very divergent based on the KSNP analysis were removed from the analysis. TN clades (defined as groups of three or more isolates that were all within 500 SNPs of each other) were identified. Next, reference-based hqSNP analyses were performed for each TN clade independently to determine high-resolution SNP differences between isolates. For the hqSNP analyses, an appropriate internal reference genome assembly (with adequate assembly quality and expected assembly size and G+C content) for each clade was identified (SRS2420927 for TN clade I, SRS2822480 for TN clade II, and SRS3010019 for TN clade III). Additionally, the Salmonella enterica subsp. enterica serovar Javiana str. CFSAN001992 assembly (GCF_000341425.1) was downloaded from the NCBI RefSeq database for use as an external and closed reference genome. The hqSNP analyses were performed, both with the internal and external references and for the 111 isolates together and for each TN clade individually. For each analysis, high quality single nucleotide polymorphisms (hqSNPs) were identified using the CFSAN SNP Pipeline v1.0.1 (Davis et al., 2015). The resulting hqSNP matrix fasta files were then used to construct phylogenetic trees as described above. The matrices were sorted and clustered using the hclust function (gtools package) in R studio. For the individual clade analyses using internal references, clusters of two or more related isolates were identified at hqSNP distance threshold levels of 5, 10 and 25; isolation date and other epidemiological information were not considered.

Genome annotation and pan-GWAS

TN isolate assemblies were annotated using Prokka v1.14-dev (Seemann, 2014) and RASTtk (Brettin et al., 2015). A pan-genome-wide association study (pan-GWAS) was performed to compare gene content among the isolates using Roary v3.12.0 (with Prokka annotation output files, previously described, used as input files) (Page et al., 2015) and statistical analysis was done using Scoary v1.6.16 (with the following arguments: -c I B BH PW EPW P -p 0.05 -e 100) (Brynildsrud et al., 2016) to identify genes or markers associated with inclusion in each TN clade. Genes predominantly present or absent among isolates in each clade were identified. Genes or clusters of genes (loci) were considered significantly associated with a clade if they had a Bonferroni-corrected P-value <0.05. From the pan-GWAS, the positively and negatively associated genes were classified by having an Odds Ratio of >1 and <1, respectively. To further analyze the results in a genomic context, loci that were adjacent or located in close proximity were combined into a single region.

In silico identification of genomic features and genes

Phage regions were identified in a diverse representative subset of the TN isolates (n = 31) using Phaster (Arndt et al., 2016; Zhou et al., 2011). Potential plasmids were predicted and classified using PlasmidFinder v2.0.2 (database version 2019-05-16) (Carattoli et al., 2014), Unicycler v0.4.8-beta (Wick et al., 2017), and plasmidSPAdes (Antipov et al., 2016) and visualized using Bandage (Wick et al., 2015). They were further confirmed by examining the associated assembly contigs for plasmid-associated genes, comparing to known plasmids using PLSDB (Galata et al., 2018) and BLASTn, and comparing coverage and G+C content to the whole assembly. Antibiotic resistance (ABR) determinants in genomes were predicted using ResFinder (90% threshold for identity and 60% for minimum length) (Zankari et al., 2012) to identify acquired ABR genes and PointFinder (90% threshold for identity and 60% for minimum length) (Zankari et al., 2017) to identify point mutations conferring ABR. The representative subset of the isolates (n = 31) was also examined for virulence factors using VirulenceFinder (Joensen et al., 2014) and VFDB VFanalyzer (Liu et al., 2018) and Salmonella Pathogenicity Islands (SPIs) using SPIFinder (v1.0; 95% threshold for identity and 60% for minimum length) (Roer et al., 2016).

Global phylogenetic analysis

The Salmonella database (Alikhan et al., 2018) on EnteroBase (Zhou et al., 2019) was queried for isolates with “human” listed as source niche in the strain metadata and “Javiana” listed as serovar in the strain metadata or experimental data (SISTR1 (Yoshida et al., 2016) or SeqSero2 (Zhang et al., 2019)). Only strains with country (and state for strains from the United States) included in the metadata were retained. A cgMLST + HierCC minimal spanning tree (RapidNJ algorithm) was created with GrapeTree (Zhou et al., 2018) on EnteroBase using all of the resulting strains. Strains that were likely not Javiana (conflicting serovar designations and distant on the tree) were removed from the dataset, leaving 466 strains (Data S2). The dataset was further refined to select representative strain(s) for each HC100 level cluster (≤100 cgMLST allelic differences). For each HC100 cluster, a single strain from each country/state was retained. If there were more than one strain from a country/state, the representative strain was chosen based on assembly quality (N50; and if N50 values were identical or similar, coverage and number of contigs were also considered). At least one TN strain representing each HC100 cluster (if available) was chosen to be included in the final dataset of genomes representing global diversity of of Salmonella ser. Javiana clinical isolates. The final dataset consisted of 162 strains: 29 TN isolates (11 from TN clade I, 3 from TN clade II, 10 from TN clade III, and the five isolates that didn’t fall into the main clades), 43 strains from other states in the US, and 90 isolates from other countries (Data S2). Collection date years ranged from 2002 to 2020. Assemblies for the non-TN strains were downloaded from Enterobase. All assemblies were analyzed using KSNP3.1 (Gardner, Slezak & Hall, 2015) and the resulting core SNP matrix fasta file was then used to construct a phylogenetic tree in Mega7 (Kumar, Stecher & Tamura, 2016) with 100 bootstrap replicates (Felsenstein, 1985). The evolutionary distances were computed using the number of differences method (Nei & Kumar, 2000) and the evolutionary history was inferred using the Neighbor-Joining method (Saitou & Nei, 1987). The final tree was visualized and annotated using iTOL (Letunic & Bork, 2016).

Comparison to polyphyletic serovars

Eight strain datasets were created: one for serovar Javiana and one each for other serovars that have been reported as polyphyletic (Derby, Kentucky, Mississippi, Montevideo, Newport, Saintpaul, and Senftenberg). The Salmonella database (Alikhan et al., 2018) on EnteroBase (Zhou et al., 2019) was queried for isolates with the specified serovar listed as serovar in the experimental data (SISTR1 (Yoshida et al., 2016) or SeqSero2 (Zhang et al., 2019)). For each, a cgMLST + HierCC minimal spanning tree (RapidNJ algorithm) was created. Strains that were likely not the serovar of interest (conflicting serovar designations and distant on the tree) were removed from the datasets. The final datasets (Data S3) were used to create cgMLST + HierCC minimal spanning trees (improved minimal spanning tree algorithm, MSTree V2) using GrapeTree (Zhou et al., 2018) on EnteroBase. The branch lengths between the cgMLST eBurstGroups (ceBGs) of the other polyphyletic serovars were used for comparison to the branch length between the two Javiana ceBGs. ceBG designations associated with each serovar were retrieved from the EnteroBase documentation (EnteroBase Team, 2018).

Results

Tennessee Salmonella ser. Javiana population structure

This analysis included a diverse set of 111 Salmonella ser. Javiana clinical isolates from TN (Data S1). On average, the assembled genomes from this study had 74.6x coverage, contained 71.7 contigs (34.8 contigs ≥ 1 kb), were 46.45 kb in length, and had 52.11% GC content (Data S1). Based on the KSNP analysis, the Salmonella ser. Javiana isolates from TN displayed a population structure with three main clades (Fig. 1). TN clade I contained 54 isolates, TN clade II contained four, TN clade III contained 48 isolates, and five isolates didn’t fall into the main clades (Fig. 1 and Table 1). Isolates in TN clades I, II, and III had average hqSNP distances of 119.4 (range 0 to 631), 210.3 (range 3 to 396), and 66 (range 0 to 361), respectively (Table 1 and Data S4).

Figure 1 Unrooted neighbor-joining KSNP tree of Tennessee clinical Salmonella ser. Javiana isolates.

Tree was constructed based on core SNPs determined by KSNP3 (Gardner, Slezak & Hall, 2015). The optimal tree with the sum of branch length of 5,916.1 is shown. TN clades I (highlighted in purple), II (green), and III (blue) are indicated. The percentage of replicate trees in which the associated taxa clustered together in the bootstrap test are indicated below branches. The tree is drawn to scale, with branch lengths (above branches) representing the number of base differences at core SNP positions per isolate (SNP distance). The analysis involved 112 isolates and 5,870 total SNP positions.

Table 1 Tennessee clade statistics and details for all isolates and individual clades, including number of isolates, hqSNP analyses details and statistics (reference genome used, mean hqSNP distance and range, and mean percent reads mapped and range), core and accessory genes as determined by Roary (Page et al., 2015), and loci associated with inclusion in each clade as determined by Scoary (Brynildsrud et al., 2016).

Clade	No. Isolates	hqSNP Analysis	Core Genome (genes)	Accessory Genome (genes)	No. of Locia Significantly Associated with Inclusion [No. genes]	No. of Significant Positively Associated Locia[No. genes]	
		Reference	Avg hqSNP distance [range]	Reads Mapped (%) [range]					
			analyzed together	analyzed separately						
All Isolates	111	GCF_000341425.1	645.3 [0-1427]	–	96.14 [90.44-98.44]	4,022	3,920	–	–	
		SRS2420927	627.5 [0-1416]	–	96.91 [90.35-99.70]					
		SRS2822480	631.6 [0-1404]	–	95.92 [90.35-99.77]					
SRS3010019	630.2 [0-1429]	–	97.59 [89.55-99.85]	
I	54	GCF_000341425.1	118.8 [0-605]	119.8 [0-618]	96.33 [90.44-98.38]	4,106	2,513	153 [338]	54 [94]	
SRS2420927	114 [0-594]	119.4 [0-631]	97.72 [90.35-99.70]	
II	4	GCF_000341425.1	193.8 [3-362]	210.3 [3-396]	97.60 [96.73-98.44]	4,290	322	22 [221]	16 [207]	
SRS2822480	189.7 [2-353]	197.5 [2-369]	99.48 [98.77-99.77]	
III	48	GCF_000341425.1	63.2 [1-366]	65.1 [0-376]	95.96 [91.41-97.65]	4,115	889	155 [332]	101 [238]	
SRS3010019	64.3 [0-367]	66.0 [0-361]	98.99 [94.19-99.85]	
Notes.

a Each locus is a single gene or group of multiple genes.

The 111 TN Salmonella ser. Javiana clinical isolates represented 47 different PFGE patterns (Fig. 2; Data S1). The most common PFGE patterns were JGGX01.0012 (n = 18), JGGX01.0065 (n = 17), and JGGX01.0072 (n = 10). TN clade I isolates represented 28 different PFGE patterns, with the most common being JGGX01.0012 (n = 18). TN clade II isolates represented three different PFGE patterns. TN clade III isolates represented 11 different PFGE patterns, with the most common being JGGX01.0065 (n = 17) and JGGX01.0072 (n = 10). Each of the five isolates that did not belong to a clade had a distinct PFGE pattern. All PFGE patterns were unique to only one TN clade.

Figure 2 Neighbor-joining hqSNP trees (Tennessee clades I, II, and III).

Trees were constructed based on hqSNPs identified by the CFSAN SNP Pipeline (Davis et al., 2015). The optimal trees are shown. The percentage of replicate trees in which the associated taxa clustered together in the bootstrap tests that are ≤0.8 are represented by branch color (maximum as green, midpoint as yellow, and minimum as red). The tree is drawn to scale, with branch lengths (above the branches) representing the number of base differences at hqSNP positions per isolate (hqSNP distance). (A) TN clade I. The sum of branch length was 1265.0. The analysis involved 54 isolates and 1261 total hqSNP positions. (B) TN clade II. The sum of branch length was 412.5. The analysis involved 4 isolates and 426 total hqSNP positions. (C) TN clade III. The sum of branch length was 794.0. The analysis involved 48 isolates and 794 total hqSNP positions.

High-quality single nucleotide polymorphism (hqSNP) analysis for cluster detection

Using a 5, 10, and 25 hqSNP distance threshold for cluster identification, 9, 12, and 10 potential clusters were identified, respectively (Data S4). The number of clusters decreases from thresholds of 10 to 25, as with the larger threshold, some of the clusters contain multiple subclusters identified at the lower threshold. Within TN clade I, five clusters were identified at each hqSNP distance thresholds of 5 and 10, and four clusters at 25 (Data S4). Only one cluster was identified at each of the distance thresholds for TN clade II (Data S4). Within TN clade III, three clusters were identified at a distance threshold of 5 hqSNPs, six clusters at 10 hqSNPs, and five clusters at 25 hqSNPs (Data S4).

To evaluate the effects of reference choice and isolate diversity, we ran our hqSNP analyses on the entire TN dataset (111 isolates) and on the three clades individually and with both internal draft assemblies and an external closed assembly as reference genomes. When all isolates were analyzed together using different reference genomes, the average hqSNP distance was lowest when using the internal reference from TN clade I and highest when using the external closed genome and the average percentage of reads mapped differed by up to 1.57% (Table 1 and Data S4). It should be noted that the closed external reference assembly we used (GCF_000341425.1) was most closely related to TN clade II in our original KSNP analysis (Fig. 1) and not representative of the overall population (at the time that this analysis was performed, there was only one closed Salmonella ser. Javiana genome available on NCBI RefSeq). For all three clades, regardless of the reference genome used, hqSNP distances were higher when they were analyzed independently, with the differences being only slightly higher for clades I and III. The average hqSNP distances for clades I and II were lower when using the internal references, but were slightly higher for TN clade III. For all three clades, the average percent reads mapped was higher when using the internal reference genome than the external reference genome, which is to be expected.

Epidemiological trends

The TN Salmonella ser. Javiana isolates were sourced from patients with an average age of 40.0 (range of 1 month to 90 years; standard deviation of 29.1). The highest incidence was in patients ≤4 (6.64 per 100,000) and ≥85 (4.16) years-old. Previous studies have reported that Salmonella ser. Javiana infections are more prevalent in infants and young children than for other serovars (Jones et al., 2008; Shaw et al., 2016; Srikantiah et al., 2004). Overall, 51.4% of isolates were from male patients (incidence of 1.73) and 46.8% from female (incidence of 1.50; Table 2). TN clades I and II contained more isolates from male patients than female, 57.4% and 75%, respectively. Conversely, 52.1% of TN clade III isolates were from female patients.

Table 2 Epidemiological data for all Tennessee isolates and individual TN clades. Percentages are in brackets.

Clade	No. of Isolates	Age	Sex	Isolation Source	Geographic Region	Isolation Month	
		<5	5–54	≥55	Male	Female	Stool	Urine	Blood	West	Middle	East	Jul.–Sep.	Other	
All Isolates	111	27 [24.3]	42 [37.8]	42 [37.8]	57 [51.4]	52 [46.8]	94 [84.7]	7 [6.3]	6 [5.4]	73 [65.8]	18 [16.2]	19 [17.1]	76 [68.5]	35 [31.5]	
I	54	15 [27.8]	23 [42.6]	16 [29.6]	31 [57.4]	22 [40.7]	50 [92.6]	0 [0.0]	3 [5.6]	32 [59.3]	8 [14.8]	14 [25.9]	36 [66.7]	18 [33.3]	
II	4	0 [0.0]	2 [50.0]	2 [50.0]	3 [75.0]	1 [25.0]	4 [100.0]	0 [0.0]	0 [0.0]	0 [0.0]	3 [75.0]	1 [25.0]	3 [75.0]	1 [25.0]	
III	48	12 [25.0]	15 [31.3]	21 [43.8]	22 [45.8]	25 [52.1]	36 [75.0]	6 [12.5]	3 [6.3]	40 [83.3]	4 [8.3]	3 [6.3]	35 [72.9]	13 [27.1]	

Most of the TN Salmonella ser. Javiana isolates were taken from stool samples (84.7%, n = 94), followed by urine (6.3%, n = 7) and blood (5.4%, n = 6) (Table 2). The portion of isolates taken from blood samples exceeds the 2.8% invasive disease outcome (defined by isolation from blood, cerebrospinal fluid, bone or joint fluid, or another sterile site; does not include urine, wound, abscess cultures) reported for this serovar in FoodNet states (Jones et al., 2008). Of the urine isolates, most (n = 6) belonged to TN clade III and all were from females, with an average patient age of 55.1 (standard deviation of 26.2). All of the isolates recovered from blood samples belonged to TN clades I (n = 3) and III (n = 3) and most were from males (n = 4), with an average patient age of 71.8 (standard deviation of 15.2). The majority of isolates recovered from extraintestinal sites were collected from elderly patients, which may indicate a correlation between invasive infection and age.

Geographical and temporal distribution

A geographical distribution can be seen for the TN isolates, with 65.8% (n = 73) isolated in counties in the western region of TN (Table 2; Data S1). In contrast, only 17.1% (n = 19) and 16.2% (n = 18) were isolated in counties in east and middle TN, respectively. Incorporating county population data, the west region had an IR of 4.69 clinical isolates per 100,000 population, while the east had an IR of 0.79 and the middle had an IR of 0.64, with an overall IR of 1.62 per 100,000 for the state (Fig. S2). Three counties had noticeably higher IR: Madison with 31.0, Crockett with 27.9, and Carroll with 25.0. As was the trend for all isolates, the majority of TN clade I and III isolates originated in the west region (59.3% and 83.3%, respectively; Table 2). However, a sizable amount of TN clade I isolates also originated in the east region (25.9%). Additionally, a temporal distribution was also clear, with 68.5% (n = 76) of isolates collected in July through September (Table 2; Data S1). This trend was also seen within the three TN clades.

Identification of mobile genetic elements (MGEs)

All of the TN isolates were examined for plasmids and 32 putative plasmids (19 unique) were identified in 30 isolates (Table 3). They ranged in size from 23 to 108 kb and included replicon types (a plasmid typing scheme based on replication control regions (Carattoli et al., 2014)) IncFIB, IncFII, IncI1, IncN3, and IncX4. In the representative subset of TN isolates (n = 31), Phaster predicted an average of 8.90 [range of 5-13] prophage regions per isolate (2.29 intact, 4.94 incomplete, and 1.68 questionable) (Data S5). The TN clade I isolates had the highest number of predicted prophage regions (average of 9.64 and range of 8–11), followed by TN clade II (average of 8.25 and range of 8–9) and TN clade III (average of 8.09 and range of 5–10).

Table 3 Putative plasmids identified in TN isolates.

Plasmid	Isolate(s) (Clade)	Replicon Type	Size (kb)	
B	SRS2628553 (I)		34.8	
	SRS2961449 (III)			
	SRS2998873 (I)			
	SRS3570561 (I)			
SRS3892739 (I)		
C	SRS2525483 (I)	IncFII (p96A)	67.9	
	SRS2547237 (I)			
	SRS2655760			
SRS3708648 (I)	
E	SRS2442409 (II)	IncFII (p14)	87.4 (47.5*)	
	SRS2822480 (II)			
	SRS3944395 (II)			
SRS3975519 (II)	
F	SRS2783476 (I)	IncI1	97.6	
SRS3932840 (I)	
I	SRS2420927 (I)	IncI1	92.7	
J	SRS3643397 (III)	IncFII	68.6	
K	SRS2442415	IncFIB (pB171)	94.9	
L	SRS2660207 (III)	IncX4	92.4	
M	SRS3453943 (I)	IncN3	54.9	
N	SRS2998915 (I)	IncI1	24.5	
P	SRS3471904 (I)		30.1	
O	SRS2998873 (I)		37.7	
A	SRS2998869 (I)		23	
SRS3721796 (III)		
D	SRS2628553 (I)		86.3	
SRS2850296 (I)		
G1	SRS2547178	IncFIB(S)		
Z	SRS2761182 (I)	IncFIB (pB171)	108	
G2	SRS3218596 (I)		46.3	
Q	SRS3742753 (I)		64	
Y	SRS2895509 (I)		23.5	

Identification of virulence factors and pathogenicity islands

All of the representative subset of TN isolates (n = 31) contained pathogenicity islands C63PI, SPI-13, and SPI-14 (Data S6). Most of the TN isolates examined contained SPI-2 (except for SRS2442409 [TN clade II] and SRS2998834 [TN clade III]) and SPI-4 (except SRS3453943 [TN clade I], SRS3643364 [TN clade III], and SRS2998834 [TN clade III]).

Most identified virulence genes were present in all of the TN isolates analyzed (Data S6), including the three genes (cdtB, pltA, and pltB) encoding the subunits of the cytolethal distending toxin (CDT) (Miller et al., 2018). Only the TN clade II isolates contained the pefC and pefD genes, which were on plasmid-associated contigs, that are part of the pef (plasmid-encoded fimbriae) operon and associated with fimbrial adherence (Bäumler et al., 1996). Some of the isolates (two from TN clade I and three that were not part of a clade) contained genes from the saf (Salmonella atypical fimbria) operon on putative plasmid-associated contigs (Folkesson et al., 1999). One isolate (SRS2442415, no clade) contained 11 genes associated with the yersiniabactin iron uptake system on a plasmid-associated contig (Carniel, 2001).

Identification of antibiotic resistance genes

All 111 TN Salmonella ser. Javiana isolates analyzed in the present study contained the aac(6′)-Iaa gene, which has been previously reported to confer aminoglycoside resistance (Shaw et al., 1993). One isolate (SRS2783476; TN clade I) contained the aph(3′)-Ia and sul3 genes on a contig (contig 32) associated with a putative plasmid. The aph(3′)-Ia gene has been found to confer resistance to aminoglycosides (Shaw et al., 1993) and sul3 gene has been shown to confer resistance to sulfonamides/sulfones through antibiotic target replacement and has been shown to be associated with resistance to sulfamethoxazole (Perreten & Boerlin, 2003).

Additionally, one isolate (SRS2628542; no clade) contained the qnrB19 gene, which has been shown to confer resistance to fluoroquinolones through physical protection of the antibiotic target (Correia et al., 2017). The qnr gene is plasmid-associated and has been linked with reduced susceptibility to ciprofloxacin (Casas et al., 2016; Crump et al., 2015; Redgrave et al., 2014). It is unclear if this gene in SRS2628542 is on a plasmid, as it is located on a small (703 bp) contig and no plasmids were predicted in this isolate. However, the gene showed 100% identity over only 72.6% of the alignment with the reference gene (accession EU432277), so it may not be functionally capable of conferring the quinolone resistance phenotype.

Clade-enriched genes

A pan-genome analysis of the TN isolates revealed that the core genome (genes contained in ≥99% of isolates) consisted of 4,022 genes and the accessory genome consisted of 3,920 genes (Table 1). The difference in gene content between the identified clades were mostly found in mobile genetic elements (91.5%).

TN clade I isolates had a core genome of 4,106 genes and an accessory genome of 2,513 genes (Table 1). This clade has a much larger accessory genome than the other two clades identified in this study. This is likely due to the large variety of mobile genetic elements (e.g., plasmids, prophages) present in isolates from this clade, which is also reflected in the much larger number of PFGE patterns (n = 28) displayed by these isolates as compared to other clades (n = 3 for TN clade II and n = 11 for TN clade III). The pan-GWAS revealed 153 loci (genes or groups of genes) to be significantly associated with inclusion in this clade (54 positively associated and 99 negatively associated) (Table 1 and Data S7). These loci consisted of 338 total genes (94 positively associated and 243 negatively associated) (Table 1). The positively associated loci are found in 29 distinct genomic regions and the majority of the genes over-represented in TN clade I (73 genes) were located in eight prophage regions (Table 4).

Table 4 Summary of Scoary results.

Summary of genome regions positively associated with inclusion in TN clades I, II, and III, as determined by Scoary (Brynildsrud et al., 2016). Loci that are adjacent or located in close proximity were combined into a single region. Full results are in Data S7.

CLADE I	CLADE II	CLADE III	
Refa	Contig	Start	Stop	MGEb	Ref	Contig	Start	Stop	MGE	Ref	Contig	Start	Stop	MGE	
a	1	105,099	105,518		c	1	4,957	5,964		d	1	39	620		
a	1	420,332	421,603	PP	c	1	385,692	385,856	PP	d	1	25,795	25,941		
a	1	699,132	699,236		c	2	47,784	63,649	MGE	d	1	106,264	106,698		
a	2	50,693	83,627	PP	c	2	74,110	92,004	PP	d	1	246,605	246,709		
a	2	98,106	98,423		c	2	130,029	149,379	PP	d	1	639,123	639,776		
a	2	235,502	235,738		c	2	191,666	192,127		d	1	727,239	767,712	MGE	
a	2	318,691	319,161		c	2	791,655	829,140	PP	d	1	1,082,149	1,118,071	PP	
a	3	208,678	208,803		c	3	150,966	168,230	MGE	d	1	1,225,448	1,226,185		
a	3	458,030	458,491		c	4	73,904	91,513	MGE	d	2	50,754	77,584	PP	
a	3	657,518	689,333	PP	c	5	66,512	66,652		d	2	77,581	100,603	MGE	
a	4	294,839	304,113	PP	c	6	134	2,179		d	2	560,052	571,035	MGE	
a	5	195,020	195,127		c	7	125,002	125,955		d	3	331,513	355,434	MGE	
a	5	235,725	236,411		c	7	182,477	183,643		d	3	492,689	492,952		
a	5	269,009	270,619		c	9	79,782	80,006		d	4	39	620		
a	5	335,429	335,854		c	9	99,100	99,297		d	4	190,030	190,365		
a	5	350,250	350,489		c	13	158	87,203	PL	d	4	190,329	190,424		
a	6	60,039	60,707		c	18	5,004	6,047		d	5	1,595	7,589	MGE	
a	6	60,750	60,974							d	5	119,382	120,038		
a	7	681	1,367							d	9	684	7,579	PP	
a	7	1,638	1,892							d	11	3,722	27,925	MGE	
a	7	197,439	205,343	PP						d	32	3	236		
a	11	33,863	33,967							e	177	12	149	MGE	
a	12	3,452	5,467	PP						f	3	470,622	471,176	MGE	
a	15	1	618							f	5	231,826	232,047		
a	15	5,734	6,108							f	9	204,729	247,594	PP	
b	1	1,642	2,379							f	14	75,550	76,125		
b	1	109,453	114,795	PP						g	1	1,076	1,198	MGE	
b	2	675,050	696,164	PP						g	4	45	617		
b	5	58,479	58,904							g	31	21	581		
Notes.

a Reference Isolate: (a) SRS2420927, (b) SRS2628565, (c) SRS2822480, (d) SRS3010019, (e) SRS3643364, (f) SRS3721796, (g) SRS3799118.

b MGE Region: Prophage (PP), Putative mobile genetic element (MGE), Plasmid (PL).

Twelve of the overrepresented genes in TN clade I correspond to pathogenicity-related protein families (as identified by PathogenFinder): DNA damage-inducible protein I, phenolic acid decarboxylase subunit D, small toxic polypeptide LdrD, PTS system fructose-specific EIIB’BC component, PTS system mannose/fructose/sorbose family IID component, prepilin peptidase dependent protein A precursor, phage DNA binding protein, and other hypothetical proteins. Overexpression of ldrD, which is part of a chromosomal toxin–antitoxin gene system (Alix & Blanc-Potard, 2009; Kawano et al., 2002), is toxic to the cell and leads to growth inhibition and rapid cell killing (Fozo et al., 2008; Kawano et al., 2002). ldrD homologs have not been found in plasmids, but may be involved in cellular response to environmental stress (Kawano et al., 2002). Prepilin peptidase dependent protein A precursor is known to be plasmid-associated (Raspoet et al., 2019; Zhang, Lory & Donnenberg, 1994) and is involved in processing of the major pilus subunit (Filloux, Michel & Bally, 1998; Zhang, Lory & Donnenberg, 1994). Other genes of interest enriched in isolates from TN clade I include those encoding autotransporter adhesin SadA, which is associated with pathogenesis, and virulence protein MsgA, which is involved with survival within macrophage (Skyberg, Logue & Nolan, 2006). Mezal, Stefanova & Khan (2013) identified msgA virulence gene in 7 (out of 50) Salmonella ser. Javiana isolates, all of which were clinical.

TN clade II isolates had a core genome of 4,290 genes and an accessory genome of 322 genes (Table 1). The pan-GWAS revealed 22 loci to be significantly associated with inclusion in this clade (16 positively associated and 6 negatively associated) (Table 1 and Data S7). These loci consisted of 221 total genes (207 positively associated and 14 negatively associated) (Table 1). The positively associated loci are found in 17 distinct genomic regions and many of the genes over-represented in TN clade II (93 genes) were located in the 87.5 kb IncFII type plasmid identified in SRS2922480 (Table 4 and Fig. S3). Among the over-represented genes contained on this plasmid are ccdAB, which are part of a toxin/antitoxin system. This system contributes to stability of the plasmid through post-segregational killing (killing new cells that do not inherit a plasmid copy during cell division) (Van Melderen, 2001). Additionally, 72 over-represented genes were located in four predicted prophage regions and 33 in three other putative MGE regions (indicated by gene annotations, clustering of genes, and/or close proximity to predicted prophage regions; Table 4). One of the overrepresented genes, alpha-xylosidase, corresponds to a pathogenicity-related protein family (as identified by PathogenFinder). VFDB identified pefC and pefD, fimbrial adherence determinants, on the plasmid (contig 13); sinH, a nonfrimbrial adherence determinant; and pipB, a TTSS-2 translocated effector.

TN clade III isolates had a core genome of 4,115 genes and an accessory genome of 889 genes (Table 1). The pan-GWAS revealed 155 loci to be significantly associated with inclusion in this clade (101 positively associated and 54 negatively associated) (Table 1 and Data S7). These loci contained 332 total genes (238 positively associated and 94 negatively associated) (Table 1). The positively associated loci are found in 29 distinct genomic regions and the majority of the genes over-represented in TN clade III were located in four predicted prophage regions (88 genes) or nine other putative MGE regions (134 genes) (Table 4). Four of the overrepresented genes correspond to pathogenicity-related protein families (as identified by PathogenFinder: arginine/lysine/ornithine decarboxylase) and other hypothetical proteins.

Global population structure

The KSNP analysis of the diverse set of global clinical Salmonella ser. Javiana strains revealed three major clades (Fig. 3; Fig. S1). Major clade I contained 107 strains, including TN isolates (from TN clades I, II, and III and the five isolates that didn’t fall into the main clades). Major clade II contained 23 strains and major clade III contained 31 strains. Strains from major clades I and II belong to the 590 cgMLST eBurstGroup (ceBG) and strains from major clade III belong to the 204 ceBG; both of these are associated with this serovar (EnteroBase Team, 2018).

Figure 3 Circular neighbor-joining KSNP tree of global clinical Salmonella ser. Javiana strains.

Tree was constructed based on core SNPs determined by KSNP3 (Gardner, Slezak & Hall, 2015). The optimal tree with the sum of branch length of 31,777.6 is shown. The percentage of replicate trees in which the associated taxa clustered together in the bootstrap test that are ≤0.8 are represented by branch color (maximum as green, midpoint as yellow, and minimum as red). The tree is drawn to scale, with branch lengths (above branches) representing the number of base differences at core SNP positions per isolate (SNP distance). The analysis involved 161 isolates and 30,657 total SNP positions. The three major clades are labeled. HC900 (ceBG) clusters are indicated (590 is not shaded and 204 is shaded in gray). TN isolates belonging to TN clades I, II, and III from our original analysis (Fig. 1) are highlighted in purple, green, and blue, respectively. A standard tree with additional metadata can be found in the supplemental files (Fig. S1).

To further explore if Salmonella ser. Javiana is polyphyletic, we constructed minimal spanning trees based on cgMLST allele distances of all available Salmonella ser. Javiana strains and of other Salmonella serovars previously described as polyphyletic (Derby, Kentucky, Mississippi, Montevideo, Newport, Saintpaul, and Senftenberg (Achtman et al., 2012; Alikhan et al., 2018; Banerji et al., 2020; Cao et al., 2013; Sangal et al., 2010; Sévellec et al., 2018; Tang et al., 2019; Timme et al., 2013; Vosik et al., 2018; Zhang, Payne & Lan, 2019)) (Fig. 4; Fig. S2). The branch length between the two ceBG clusters on the ser. Javiana tree was 1,423 allelic differences. The branch lengths between ceBG clusters on the other trees ranged from 2,280-2,769 allelic differences for ser. Derby, 2,452-2,756 for ser. Kentucky, 929-2,850 for ser. Mississippi, 1,188-2,844 for ser. Montevideo, 957-2,636 for ser. Newport, 1,513-2,686 for ser. Saintpaul, and 1,844-2,743 for ser. Senftenberg.

Figure 4 Minimal spanning trees of Salmonella. ser. Javiana and polyphyletic serovar strains.

Minimal spanning trees were constructed from cgMLST data using GrapeTree on Enterobase with the improved minimal spanning tree algorithm (MSTree V2). Separate trees were constructed for each (A) Salmonella ser. Javiana, (B) ser. Mississippi, (C) ser. Montevideo, and (D) ser. Newport. Nodes are color coded by ceBG designations (see legend in each panel) and ceBG designations associated with each serovar are in red boxes. Branch lengths (representing cgMLST allelic differences) are shown in red above branches between ceBG clusters. Trees for other serovars are in Fig. S2.

Discussion

The main objective of this study was to use WGS data to retrospectively examine the population structure of Salmonella ser. Javiana, both from a local (state of TN) and global perspective. The phylogenetic analysis of the 111 Salmonella ser. Javiana isolates from TN revealed a population structure with three main clades, with the majority of isolates found in TN clades I and III. Research published on the population structure of this serovar is limited. One comparable study, conducted by Mezal, Stefanova & Khan (2013) used PFGE to assess the relatedness of 50 Salmonella ser. Javiana isolates from food, environmental, and clinical sources. They found that the isolates represented 34 distinct PFGE patterns and grouped into five clusters of two or more isolates; the 30 clinical isolates represented 23 distinct PFGE patterns (compared to the 111 TN clinical isolates in the present study representing 47 distinct PFGE patterns) and spanned all five clusters. The diversity of PFGE patterns suggested that differences in genome content between Salmonella ser. Javiana isolates are common. In this study, we found that differences in gene content between the TN clades were mostly attributed to mobile genetic elements (e.g., prophage regions and plasmids), with TN clade I exhibiting the highest level of accessory genome diversity.

The phylogenetic analysis of the diverse set of global clinical Salmonella ser. Javiana strains revealed three major clades (Fig. 3; Fig. S1). Major clade I contained most of the strains, including all of the TN isolates. This indicates that the population of this serovar in TN represents only a portion of the global genomic diversity. Strains from major clades I and II belong to the 590 cgMLST eBurstGroup (ceBG) and strains from major clade III belong to the 204 ceBG. ceBGs are equivalent to eBurstGroups (eBGs; in legacy 7-gene MLST), which have been shown to correspond to serovar designations (EnteroBase Team, 2018; Zhou et al., 2019). Typically, monophyletic serovar isolates will belong to a single eBG, while polyphyletic serovar isolates will belong to multiple eBGs (Achtman et al., 2012; Alikhan et al., 2018; Banerji et al., 2020).The observation that the global clinical Salmonella ser. Javiana isolates consisted of multiple major clusters and two ceBGs suggests that this serovar may be polyphyletic. Ashton et al. (2016) characterized serovars in lineage 3 of S. enterica subspecies I (which includes serovars Bredeney, Chester, Javiana, Montevideo, Oranienburg, and Poona) as polyphyletic and containing multiple eBGs. The branch length between the two ceBG clusters on the ser. Javiana tree (1,423 allelic differences) was comparable to the branch lengths between ceBGs on the other serovar trees (929-2,850 allelic differences) (Fig. 4 and Fig. S2). Based on this comparison, Salmonella ser. Javiana may be considered a polyphyletic serovar, although this depends on the branch length cutoff that is applied.

As WGS is becoming more commonly used for public health applications (e.g., cluster detection and outbreak investigation), it is important to understand genomic population structure of surveilled disease-causing microorganisms, specifically at the serovar level for Salmonella. Genomic distance thresholds (based on hqSNP or allelic distances) are an important factor used for identifying potential disease clusters of public health importance, but other factors are typically considered, including isolation date, number of isolates, and epidemiological data. In the present study, we found that using different hqSNP distance thresholds for cluster identification resulted in different numbers of potential clusters and associated isolates (Data S4). The selected threshold for cluster detection should be empirically determined so that it is larger than typical inter-genomic distances between outbreak strains, but smaller than typical inter-genomic distances between outbreak and background (non-outbreak) isolates. Inter-genomic SNP distances among Salmonella outbreak strains are typically small (in the 2 to 12 SNP range), but in some cases can be quite large (up to 249 SNPs) and likely vary from serovar to serovar (Leekitcharoenphon et al., 2014). Isolates from zoonotic or prolonged (e.g., persistent contamination from production environments) outbreaks will likely have larger genomic distances and outbreaks with very large genomic distances are typically polyclonal events (Besser et al., 2019). For Salmonella, the CDC uses a working definition of ≥3 cases within a 60-day period with ≤10 cgMLST allele differences, with ∼2 cases that have ≤5 allele differences (Besser et al., 2019). Thresholds can have impacts on epidemiological investigations; if they are set too low, isolates belonging to the same outbreak event may be mistakenly excluded from the cluster or separated into different clusters and, if they are set too high, background isolates may be inadvertently included in the cluster, making epidemiological investigations difficult, particularly source attribution. In the present study, as the hqSNP distance threshold was increased, the number of included isolates also increased. Increases in numbers and/or sizes of potential clusters may impact the ability of public health departments to further investigate them due to resource constraints. Thresholds may also need to be adjusted based on the timeline of the suspected outbreak (lower for short-term and higher for prolonged outbreaks). As we move forward with using WGS for routine surveillance and cluster detection of this serovar, more clusters may be successfully detected and investigated. In turn, this will provide information on typical genomic distances that can be used to establish and evaluate an appropriate serovar-specific threshold for cluster detection.

Another important consideration when using hqSNP calling analyses for epidemiological cluster detection is whether polyphyletic serovars or those with genetically diverse clades should be analyzed together or if each clade should be analyzed independently. An additional consideration is the choice of reference genome. These choices can affect the percentage of reads mapped to the reference genome and, in turn, the results of the analysis (primarily, hqSNP distances). Better performance (i.e., higher read mapping) would be expected when using closed genomes as references for hqSNP calling. However, some research has shown that using closed vs draft genomes as references have limited impact on hqSNP calling phylogeny reconstruction (Jagadeesan et al., 2019; Portmann et al., 2018). In the current study, we still achieved a level of high-quality mapping (>95%, as recommended by Katz et al., 2017; Table 1) when using draft genomes as references. As these types of studies are performed, representatives from each clade should be selected for long-read sequencing to establish high-quality reference genomes that can be used to further evaluate hqSNP distances. Additionally, when analyzing the TN isolates together or each clade separately and with internal or external reference genomes, similar levels of performance were achieved. This is likely due to the lack of diversity in the core genome of the isolates and the fact that the majority of the gene content differences between isolates from each clade were attributed to MGEs (plasmids and prophage regions). Commonly used hqSNP pipelines filter out SNPs that are found in close proximity and/or mask phage regions (Katz et al., 2017; Strain et al., 2019) and only SNPs present in genomic regions shared between isolates and the reference genome are identified in the analysis. When viewed from a global context, the TN isolates were all part of a single major clade and a single ceBG, which may also explain the similar levels of performance seen with the different analysis strategies.

Notable geographical and temporal patterns were observed for the Salmonella ser. Javiana isolates from TN. The geographical distribution within the state (most isolates from patients in counties in the western region; Table 2 and Fig. S3) is consistent with other reported data (Centers for Disease Control and Prevention, 2013; Mukherjee et al., 2020). This geographical distribution may be associated with the higher percentage of fresh forested/scrub-shrub wetlands in these west TN counties (Huang et al., 2017). A similar geographical distribution has been described in GA, with Salmonella ser. Javiana cases occurring more frequently in the southern part of state (Clarkson et al., 2010). Despite this, Harris et al. (2018) were unable to isolate Salmonella ser. Javiana from storm runoff or irrigation ponds used by fresh produce growers in South Georgia even though this is a high incidence area. The temporal distribution (most isolates collected July-September; Table 2) is in accordance with the notable seasonality of this serovar reported elsewhere (Clarkson et al., 2010; Srikantiah et al., 2004).

Salmonella virulence factors aid in host colonization and pathogenicity by assisting the pathogen in attaching to, invading, and replicating within host cells, intra-and extracellular survival, evading host defenses, and outcompeting the gut microbiome and include adhesion systems, capsule, flagella, and toxins (Jajere, 2019). Virulence factors and related genes are frequently clustered together in pathogenicity islands, which are often found on mobile genetic elements (MGEs), such as plasmids and prophages (Cheng, Eade & Wiedmann, 2019; Jacobsen et al., 2011). Eight Salmonella Pathogenicity Islands (SPIs) or islets (SPI-1, SPI-2, SPI-4, SPI-5, SPI-9, SPI-11, SPI-12, and CS54) are commonly found in most non-typhoidal serovars (Den Bakker et al., 2011; Jacobsen et al., 2011). All of the representative subset of TN isolates analyzed for SPIs contained C63PI, SPI-13, and SPI-14 (Data S6). C63PI, which is located within SPI-1, contains the sit operon that encodes an iron uptake system (Schmidt & Hensel, 2004; Zhou, Hardt & Galán, 1999). SPI-13 has been associated with macrophage internalization and virulence in chickens and mice (Cheng, Eade & Wiedmann, 2019; Elder et al., 2016; Espinoza et al., 2017; Shah et al., 2005). SPI-14 is involved in epithelial invasion and pathogenicity in chickens (Cheng, Eade & Wiedmann, 2019; Fookes et al., 2011; Shah et al., 2005). Most of the representative subset of TN isolates analyzed for SPI contained SPI-2 and SPI-4. SPI-2 encodes a type III secretion system (T3SS-2), which is involved in intracellular survival and replication, immune evasion, and systemic pathogenicity (Schmidt & Hensel, 2004; Tsai & Coombes, 2019), replication within macrophages, and systemic infections (Jajere, 2019). SPI-4 encodes genes for toxin secretion and apoptosis and is involved in intracellular (macrophage) survival (Jajere, 2019). All three genes associated with the cytolethal distending toxin were identified in all of the representative subset of TN isolates, which is in agreement with other studies that have identified these three genes in all Salmonella ser. Javiana isolates tested (Mezal, Stefanova & Khan, 2013; Miller & Wiedmann, 2016). Other virulence genes that differed among isolates were mainly associated with mobile genetic elements.

All 111 TN Salmonella ser. Javiana isolates analyzed in the present study contained the aac(6′)-Iaa gene, which is associated with aminoglycoside resistance (Shaw et al., 1993). However, there is evidence that this gene is cryptic and no longer confers phenotypic aminoglycoside resistance (Leon et al., 2018; Salipante & Hall, 2003), which is consistent with the low prevalence of phenotypic resistance to amikacin and gentamicin (0.04%) in U.S. clinical Salmonella ser. Javiana isolates (Centers for Disease Control and Prevention, 2019b). This finding highlights the complexity of antimicrobial resistance. The three other antibiotic resistance genes identified in this study (aph(3′)-Ia, sul3, and qnrB19) were each only present in a single isolate. The low prevalence of these three genes is consistent with the low prevalence of phenotypic resistance to gentamicin and kanamycin (0.12%), sulfamethoxazole/sulfisoxazole (0.63%), trimethoprim-sulfamethoxazole (0.21%), and ciprofloxacin (0%) seen in U.S. clinical Salmonella ser. Javiana isolates (Centers for Disease Control and Prevention, 2019b). Additionally, the hypothesis that the qnrB19 gene may not be functional is further supported by the fact that phenotypic ciprofloxacin resistance has not been reported in Salmonella ser. Javiana clinical isolates (Centers for Disease Control and Prevention, 2019b). As aminoglycosides are not typically used to treat Salmonella infections, the presence of the aac(6′)-Iaa and aph(3′)-Ia genes is of little clinical significance. Overall, these data show a low prevalence of genes associated antibiotic resistance in Salmonella ser. Javiana from TN. However, antibiotic susceptibility testing would need to be performed on these isolates to confirm.

Conclusions

This study demonstrates the population structure of Salmonella ser. Javiana in Tennessee and globally. As this is a clinically important Salmonella serovar, understanding the phylogeny can provide guidance for phylogenetic analyses and cluster detection for public health surveillance and response. We show that Salmonella ser. Javiana clinical isolates from TN show geospatial and temporal distribution, with most isolates originating from the western part of the state and during the summer months (July, August, and September). Based on the results of the pan-GWAS, it is clear that MGEs (namely plasmids and prophage regions) in the genome account for most of the differences in gene content between the three main clades of this serovar. This is noteworthy, as clinically-relevant genes (like ABR-conferring or virulence-related genes) can be found in these regions and they could potentially be used for isolate characterization. Additionally, we found that when performing hqSNP analysis for epidemiological cluster detection with the TN isolates, it is not necessary to first divide the isolates into clades, as we found this only minimally increases the SNP differences between isolates; however the TN isolates all belonged to a single global major clade and single ceBG, so this may only be applicable to less diverse populations. Further research should include clinical Salmonella ser. Javiana isolates and associated metadata from other states to obtain a more complete representation of the population structure of and epidemiological information about Salmonella ser. Javiana in the United States and an analysis of disease severity and gene content could assist in the identification of genes that may be involved in virulence. Another research direction would be to include isolates from other sources (e.g., environmental, animal, food) in a phylogenetic analysis, which may expand our understanding of the population structure and including isolates with diverse isolation sources may provide insight into source attribution and potential recommendations to prevent morbidity.

Supplemental Information

Figure S1 Neighbor-joining KSNP tree of global clinical Salmonella ser. Javiana strains

Tree was constructed based on core SNPs determined by KSNP3 (Gardner, Slezak & Hall, 2015). The optimal tree with the sum of branch length of 31,777.6 is shown. The percentage of replicate trees in which the associated taxa clustered together in the bootstrap test are represented by branch color (maximum as green, midpoint as yellow, and minimum as red). The tree is drawn to scale, with branch lengths (above branches) representing the number of base differences at core SNP positions per isolate (SNP distance). The analysis involved 161 isolates and 30,657 total SNP positions. The three major clades are labeled. HC900 (ceBG) clusters are indicated (590 is not shaded and 204 is shaded in gray). TN isolates belonging to TN clades I, II, and III from our original analysis (Fig. 1) are highlighted in purple, green, and blue, respectively. Metadata, including HC100 cluster designations, country, region/state, and collection year are listed to the right of node labels.

Click here for additional data file.

Figure S2 Minimal spanning trees of Salmonella ser. Javiana and polyphyletic serovar strains

Minimal spanning trees were constructed from cgMLST data using GrapeTree on Enterobase with the improved minimal spanning tree algorithm (MSTree V2). Separate trees were constructed for each (A) Salmonella ser. Javiana, (B) ser. Derby, (C) ser. Kentucky, (D) ser. Mississippi, (E) ser. Montevideo, (F) ser. Newport, (G) ser. Saintpaul, and (H) ser. Senftenberg. Nodes are color coded by ceBG designations (see legend in each panel) and ceBG designations associated with each serovar are in red boxes. Branch lengths (representing cgMLST allelic differences) are shown in red above branches between ceBG clusters.

Click here for additional data file.

Figure S3 Map of incidence rates

Click here for additional data file.

Figure S4 Tennessee clade II plasmid

Circular representation of the 87.5 kb found in TN clade II isolate SRS2822480. Genes in green signify over-represented Clade II genes and their corresponding Prokka annotations. RASTtk annotations that differ from the Prokka annotations are in gray. Figure was created using Geneious Prime 2019.1.1 ( https://www.geneious.com ).

Click here for additional data file.

Data S1 Tennessee isolate details, IDs, and Assembly Statistics

Click here for additional data file.

Data S2 Global Salmonella ser. Javiana Clinical Strains Dataset

Strain dataset used to evaluate the global population structure of Salmonella ser. Javiana clinical isolates. The original dataset included 526 isolates and the final dataset consisted of 161 isolates (indicated in the third column from the right). Includes strain identifiers, metadata (e.g., source, location), experimental data (e.g., in silico serotyping, MLST, cgMLST, wgMLST, and assembly stats), TN clade (for TN isolates included in the first KSNP analysis), and lineage (determined from the KSNP analysis of this dataset).

Click here for additional data file.

Data S3 Polyphyletic Serovar Datasets

Isolate datasets used to construct minimal spanning trees for the following serovars: (A) Javiana, (B) Derby, (C) Kentucky, (D) Mississippi, (E) Montevideo, (F) Newport, (G) Saintpaul, and (H) Senftenberg. Includes strain identifiers, metadata (e.g., source, location), and experimental data (e.g., in silico serotyping, MLST, cgMLST, wgMLST, and assembly stats). Datasets are also publicly available on EnteroBase.

Click here for additional data file.

Data S4 Clustered hqSNP Distance Matrix Heatmaps (TN Clades I, II, and III and all isolates)

Clustered matrices contain hqSNP distances between isolates as determined by the CFSAN SNP Pipeline (Davis et al., 2015) . The seven tabs (A, B, B2, C, C2, D, and D2) contain the results from when isolates from when all isolates were analyzed together with the external reference or the three internal references and the last six tabs (E, F, G, H, I, and J) from when the clades were analyzed independently with the external reference or the respective internal reference. Cells are colored to reflect hqSNP distances (low are white, high are dark red). Potential clusters at various hqSNP distance thresholds are indicated (£25 hqSNPs with dashed lines, ≤10 hqSNPs with dotted lines, and £5 hqSNPs with solid lines).

Click here for additional data file.

Data S5 Prophage regions present in the representative subset of TN isolates, as determined by Phaster (Arndt et al., 2016; Zhou et al., 2011)

Click here for additional data file.

Data S6 Virulence Factors and Salmonella Pathogenicity Islands (PAIs)

(A) Virulence factors determined by VFanalyzer and (B) Salmonella pathogenicity islands determined by SPIfinder.

Click here for additional data file.

Data S7 Scoary Results (full)

Click here for additional data file.

Additional Information and Declarations

Competing Interests

Author Contributions

DNA Deposition

Data Availability

The authors declare there are no competing interests.

Lauren K. Hudson conceived and designed the experiments, performed the experiments, analyzed the data, prepared figures and/or tables, authored or reviewed drafts of the paper, and approved the final draft.

Lisha Constantine-Renna performed the experiments, prepared figures and/or tables, authored or reviewed drafts of the paper, and approved the final draft.

Linda Thomas, Christina Moore and Xiaorong Qian performed the experiments, authored or reviewed drafts of the paper, and approved the final draft.

Katie Garman, John R. Dunn and Thomas G. Denes conceived and designed the experiments, authored or reviewed drafts of the paper, and approved the final draft.

The following information was supplied regarding the deposition of DNA sequences:

All 111 S. Javiana reads are available at NCBI SRA:

SRS2420927 SRS2525483, SRS2547237, SRS2547417, SRS2609391,

SRS2628553, SRS2628565, SRS2660215, SRS2761182, SRS2761294,

SRS2783476, SRS2845130, SRS2850296, SRS2895509, SRS2922972,

SRS2938622, SRS2938629, SRS2938662, SRS2998847, SRS2998869,

SRS2998873, SRS2998915, SRS3007164, SRS3007170, SRS3080274,

SRS3080280, SRS3218596, SRS3384774, SRS3453937, SRS3453943,

SRS3471848, SRS3471904, SRS3531727, SRS3531728, SRS3570561,

SRS3614303, SRS3614304, SRS3643366, SRS3643394, SRS3703451,

SRS3703452, SRS3703470, SRS3708648, SRS3721753, SRS3742747,

SRS3742753, SRS3855333, SRS3871671, SRS3871678, SRS3876540,

SRS3877810, SRS3892739, SRS3932748, SRS3932840, SRS2442409,

SRS2822480, SRS3944395, SRS3975519, SRS2420915, SRS2420931,

SRS2532115, SRS2609377, SRS2609378, SRS2609380, SRS2609382,

SRS2660207, SRS2695323, SRS2761300, SRS2845135, SRS2850289,

SRS2926162, SRS2938625, SRS2938638, SRS2961449, SRS2998834,

SRS2998855, SRS2998913, SRS3010019, SRS3021992, SRS3021995,

SRS3080188, SRS3242649, SRS3277971, SRS3597673, SRS3623378,

SRS3623430, SRS3643364, SRS3643397, SRS3643408, SRS3676310,

SRS3703468, SRS3721793, SRS3721795, SRS3721796, SRS3737739,

SRS3753120, SRS3799118, SRS3871596, SRS3871610, SRS3876542,

SRS3877801, SRS3877812, SRS3932734, SRS3944382, SRS3950101,

SRS3975507, SRS2442415, SRS2547178, SRS2628542, SRS2655760,

SRS2709963.

The following information was supplied regarding data availability:

The data is available in the Supplemental Files.

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
