# Peer review of "Genomic characterization and phylogenetic analysis of Salmonella enterica serovar Javiana"

_PeerJ, doi:10.7717/peerj.10256_

## Round 0.1 · original submission · Major Revisions

The reviewers generally liked your manuscript, but asked for some changes. I agree that comparing your findings with a wider sample of other publicly available genomes, e.g. on Enterobase, would render results more robust.

·

Basic reporting

The manuscript is written with very good scientific English. It is structured clearly. The Background the authors provide for the investigation is very thorough with adequate references. The manuscript is self-contained with the results that are relevant to the objectives. The number of the figures included in the manuscript itself is appropriate. I think Table 3 could be included in the supplemental material and table 4 is not needed since the same information is included in the supplemental table 4.

The following edits should be made:
1. Line 21 and throughout the manuscript: S. Javiana is not correct Salmonella nomenclature. The genus name Salmonella cannot be shortened as “S.” without spelling out the species name enterica. So after first mentioning and spelling out the entire genus-species-serovar name, the following references to it are acceptable moving forward: “S. enterica serovar Javiana”, “Salmonella ser. Javiana or “ser. Javiana”. Please check the Pasteur Institute's guidance at: https://www.pasteur.fr/sites/default/files/veng_0.pdf.
2. Line 107: delete the repeated “cucumbers”.
3. Line 165, Supplemental table 1: for each strain, please indicate whether it was part of a known outbreak or considered sporadic.
4. Line 242: “multiple sub-clusters” instead of “multiple of the clusters”.
5. Line 243: “potential disease clusters of public health importance” instead of “potential clusters in public health”.
6. Line 277: comma after “blood”.
7. Line 303: “of” after the percentage.
8. Line 360: ‘is in agreement with” instead of “is comparable to”.
9. Line 369: ‘it” instead of “this”.

Experimental design

Research questions are well defined and fill a gap in the current state of knowledge. AS for the methods I have the following comments:
1. Lines 163-176: include a brief description (library prep method and sequencing chemistry at a minimum) of the sequencing methods.

2. Lines 178-198: please clarify what was the reasoning for using the two different SNP discovery methods in this study: the reference-free KSNP and the reference-based CFSAN pipeline? You state that you used the core SNP matrices from the KSNP to evaluate the population structure, not matrices based on all (core and ancillary) SNPs and given that the CFSAN pipeline removes mobile elements from hqSNP calling, the results should be expected to be close the same from the two pipelines. If the reason was that you were concerned the reference-based analysis would bias the population structure then that should be clearly stated over here.

3. Lines 187-192: the choice of the hqSNP reference sequence for each lineage: typically you want the reference sequence to be closely related to the study population, so what exactly do the authors mean when they state that one criterium for the appropriate reference sequence was that is was “not closely related to other isolates”? Also what was the rationale of using internal references for clades I and III but an external reference for clade II?

4. Lines 223-227: what was the reason for only using the 3 reference sequences to identify the virulence factors? Given, how poorly understood the virulence markers currently are in Salmonella, I would determine the virulence factors either in all sequences or at least in a subset of strains from each clade to get an idea how much strain level variation there is within the clade.

Validity of the findings

For the findings and conclusions I have the following comments/suggestions:
1. Lines 275-284: the sample source being extra-intestinal may be a poor indication of severe disease. The number of hospitalizations and deaths would be better. TN is a FoodNet site, so the disease outcome information should be available for these isolates and should be included in Table 2 and it should be discussed whether severe outcome was overrepresented in any of the clades.

2. Lines 445-448: what was the reasoning to use the clade II reference as a reference when analyzing all isolates together? Because it was closed and the clade I and III references were not? Clade II appears to be the least representative of the overall Javiana population. I would like to see this analysis being performed also using the clade I and III references. As you state on lines 459-462, the use of closed vs. draft reference should not have a drastic impact and sometimes using a draft reference that is more closely related to the study population actually gives better results than using a more distantly related closed reference. In the case of Javiana, the choice of reference may not make much difference because it looks like there may not be that much diversity in the core genome, but the authors should at least attempt to prove that by performing the overall analysis of the 111 sequences using also the clade I and III references.

3. Lines 475-477: this is an over-statement unless the overall analysis is also performed using the clade I and the clade III references to prove that for Javiana the choice of reference does not make much difference.

4. Lines 481-483: more importantly, a larger dataset, particularly from other FoodNet sites where the information about disease severity (hospitalizations, death) is known would help shed light on the role of potential virulence genes.

Additional comments

No additional comments

Reviewer 2 ·

Basic reporting

The basic reporting is excellent.

Experimental design

The manuscript is well written and clear. The Introduction is well presented, highlights the relevant aspects of why the research was undertaken, and provides an appropriate, easy to read introduction. It highlights that S. Javiana is a highly-prevalent enteric food pathogen of concern, and discusses its potential as a carrier of antimicrobial resistance. The authors are correct in their assertion that SJ’s population structure is important to study.
The Materials and Methods are well presented and logical, and they use well-respected bioinformatics tools in the appropriate contexts. On Line 170: I recommend that they make it clear that (ILLUMINACLIP:etc are parameters passed for Trimmomatic to use.
The Results & Discussion are also clearly presented, and I found all of the Figure, Tables and Supplementary information appropriately tagged and in the appropriate places. The authors present a well supported case that 106 of the 111 SJ genomes that were sequenced fall into three discrete clades. If they were to suggest that and leave it there, I’d have accepted the manuscript pretty quickly without major revisions.

Validity of the findings

I have huge doubts that selecting, sequencing and analysing 110 strains from Tennessee, collected over 18 months, is sufficient to then declare that S. Javiana is polyphyletic. I suggest that the authors provide better context especially with regard to the worldwide distribution of SJ – which is readily available. I would like the authors to put their analyses in the context of many more genomes of SJ, including those found elsewhere in the US as well as those found internationally. These genomes are readily available.
Furthermore, I’d like to see these analyses presented in the context of a global analysis of Salmonella genomes – most of which are in Enterobase, to see if the branch lengths of their proposed polyphyletic clades bear similar relationships to other polyphyletic serotypes of Salmonella e.g. Newport, Java, etc. Even a simple minimum spanning tree would be sufficient.
Please perform those requested analyses, and add them to the manuscript, adjusting the conclusions to accommodate the enlarged analysis, OR, rewrite the manuscript removing the assertion that SJ is polyphyletic and replace it with the finding that it has three highly diverse clades.

Additional comments

Other than my above comments, it is a highly polished manuscript and a pleasure to read.

---

## Round 0.2 · Minor Revisions

Please address the few remaining suggestions for changes by the reviewer.

·

Basic reporting

No comment

Experimental design

The authors have adequately addressed my concerns in the revision.

Validity of the findings

1. Lines 545-560: SNP distances in an epidemiologically verified outbreak that exceed > 25-30 SNPs typically indicate the presence of polyclonal contamination, i.e. multiple different strains causing an outbreak. An outbreak with >200 plus SNPs certainly is a polyclonal one and should be noted here if the authors wish to use this reference. Cluster detection thresholds really don’t apply to polyclonal outbreaks which are challenging to investigate no matter what typing method is used. A vast majority of the Salmonella outbreaks are within 5-20 SNP range, the upper range often seen in zoonotic outbreaks or long lasting outbreaks involving persistent environmental contamination. An article discussing outbreak ecology for enteric organisms is here: Besser, J.M., Carleton, H.A., Trees, E., Stroika, S.G., Hise, K., Wise, M., Gerner-Smidt, P. (2019) Interpretation of Whole-Genome Sequencing for Enteric Disease Surveillance and Outbreak Investigation. Foodborne Path. Dis. 16. Published Online:9 Jul 2019https://doi.org/10.1089/fpd.2019.2636
2. Lines 574-581: also when put in the global context, the TN strains all belonged to the same eBurst group and clade. So while three clades appeared to be separating in the local analysis containing just the TN strains, from the global context the TN strains are monophyletic which may explain the equal performance of the different references.

Additional comments

1. Line 340: add “age’ after “an average”
2. Line 511: “are common” instead of “is common”
3. Line 617: “in all of the representative…”
4. Line 658: clarify that the TN isolates belong to a single global lineage and single eBurst group

---

## Round 0.3 · accepted · Accept

All comments have now been addressed.